# Evaluation of the Efficacy and Safety of an Exercise Program for Persons with Total Hip or Total Knee Replacement: Study Protocol for a Randomized Controlled Trial

**DOI:** 10.3390/ijerph18136732

**Published:** 2021-06-23

**Authors:** Giuseppe Barone, Raffaele Zinno, Erika Pinelli, Francesco Benvenuti, Laura Bragonzoni

**Affiliations:** Department of Life Quality Studies, Campus Rimini, University of Bologna, Corso d’Augusto, 237, 47921 Rimini, Italy; giuseppe.barone8@unibo.it (G.B.); raffaele.zinno2@unibo.it (R.Z.); benve.francis@gmail.com (F.B.); laura.bragonzoni4@unibo.it (L.B.)

**Keywords:** physical activity, osteoarthritis, quality of life, replacement

## Abstract

Total hip replacement (THR) and total knee replacement (TKR) are among the most common elective surgical procedures. There is a large consensus on the importance of physical activity promotion for an active lifestyle in persons who underwent THR or TKR to prevent or mitigate disability and improve the quality of life (QoL) in the long term. However, there is no best practice in exercise and physical activity specifically designed for these persons. The present protocol aims to evaluate the efficacy and safety of an exercise program (6 month duration) designed for improving quality of life in people who had undergone THR or TKR. This paper describes a randomized controlled trial protocol that involves persons with THR or TKR. The participant will be randomly assigned to an intervention group or a control group. The intervention group will perform post-rehabilitation supervised training; the control group will be requested to follow the usual care. The primary outcome is QoL, measured with the Short-Form Health Survey (SF-36); Secondary outcomes are clinical, functional and lifestyle measures that may influence QoL. The results of this study could provide evidence for clinicians, exercise trainers, and policymakers toward a strategy that ensures safe and effective exercise physical activity after surgery.

## 1. Introduction

Total hip replacement (THR) and total knee replacement (TKR) are among the most common elective surgical procedures [1,2,3,4]. Osteoarthritis is the most common underlying condition for both THR and TKR. A large body of published evidence has consistently shown that these surgical procedures successfully decrease pain and improve mobility and quality of life [5,6].

The utilization rates of THR and TKR procedures have been increasing in the last 2–3 decades due to technical surgical improvements, significant benefits obtained, and the populations increased longevity. This volume growth poses an increasing economic burden in healthcare systems in terms of hospitalization costs and subsequent rehabilitation [7,8]. This trend is similar in all developed countries, although the difference in these surgical procedures’ incidence may be due to variations in economic status, health care delivery systems, patient preferences, or osteoarthritis prevalence [9].

As THR and TKR relieve joint pain, this represents an opportunity for these individuals to become more physically active [10]. Regular physical activity is one of the most effective interventions to improve the prevalent chronic comorbid conditions, including obesity, diabetes, hypertension, and cardiovascular deconditioning that commonly coexist with hip and knee osteoarthritis [11]. Furthermore, as physical activity induces quadriceps hypertrophy and improves strength, it is one of the main modifiable factors in patients with knee and hip osteoarthrosis who often exhibit weakness and atrophy [12] Therefore, THR and TKR could benefit not only the overall health of the individual undergoing surgery [13,14], but also relieve symptoms by increasing physical activity.

However, individuals who have completed THR or TKR treatment (surgery plus subsequent rehabilitation phase) do not seem to increase physical activity from pre-to-post-surgery. American [15,16] and Dutch [17,18] studies, show that individuals who undergo THR or TKR tend to be older sedentary adults, and it is estimated that around 49% of them are over-weight or obese, 16% have diabetes, and about half have high blood pressure. Studies on physical activity changes due to THR or TKR are challenged by wide variability in demographics, methods used to assess physical activity, and different care pathways used across studies [13,14]. However, the studies that assessed PA from pre- to post-THR and -TKR have reported that physical activity does not change the first three months post-surgery. The results of studies with follow-up assessments longer than three months but shorter than 12 months are contradictory, and the results of follow-ups longer than 12 months provide weak evidence of increased physical activity [13,14,17,18,19]. Vissers et al. [20] reported that 4 year post-surgery patients continued to improve their perceived physical functioning, capacity, and performance of daily life activities. However, Hootman et al. [21] and Wallis et al. [22] found that persons who received THR or TKR are significantly less likely to meet WHO physical activity guidelines [23]. In addition, only 48% individuals after hip and 60% after knee replacement perform more than 7000 steps per day, respectively [21,22].

To date, there are limited data to guide orthopedic surgeons’ recommendations regarding leisure time, exercises, and sports activities after THR or TKR. Moreover, there is little evidence on the strategies to improve and maintain physical function after rehabilitation in the long term. However, it is well known that these patients have significant muscular atrophy and weakness in the affected limb that can persist for months after surgery [24], with mobility deficits remaining for many years [13,14]. Exercise programs beyond the initial post-operative rehabilitation period have been shown to reduce pain and joint stiffness, improve physical function and lessen the chance of accidental falls after surgery [25,26,27,28,29,30]. These programs have generally used strengthening exercises and functional tasks such as stair climbing to improve muscular strength and power, walking speed, and mobility. Participating in these exercise programs, persons can be physically prepared to become active again in physical activity and sport. However, a disadvantage of these programs is that persons need to exercise under clinicians’ supervision at a hospital or rehabilitation center. This makes the delivery of these programs expensive due to the high costs associated with supervised treatment and transport. In addition, some persons are excluded because of difficulties with mobility and transport to a center preclude participation [27].

Current recommendations of health-enhancing sports activities in the long term are very generic and are based on the orthopedic surgeons’ opinions [31,32,33]. The following points for exercise prescription have been evidenced. Patients should be encouraged to remain physically active to improve general health, and bone mineral density. There is evidence that increased bone quality will improve prosthesis fixation and decrease the incidence of early loosening [34,35]. Factors such as wear, joint load, intensity, and type of prosthesis must be taken into account for each subject and sport to safely recommend a specific activity after THR or TKR. It has been shown that the wear decrease is one of the main factors in improving long-term results after total joint replacement. Wear is dependent on the load, the number of steps and the material properties of joint replacements [34,35]. It is unwise to start technically demanding activities after THR or TKR, as the joint loads and the risk for injuries are generally higher for these activities in unskilled individuals [32]. Finally, it is essential to distinguish among suitable activities following THR or TKR. In order to recommend suitable physical activity after total knee replacement, it is important to consider both the load and the knee flexion angle of the peak load, while for total hip replacement, which involves a ball and socket joint, the flexion angle does not play an important role. Thus, it is prudent to be more conservative after total knee arthroplasty than after total hip arthroplasty for activities that exhibit high joint loads in knee flexion [32]. 

In conclusion, although there is a large consensus on the importance of physical function improvement to prevent or mitigate disability in persons who underwent THR/TKR in the long term, there is no best practice specifically designed to these persons. Therefore, taking into account previously published experiences [25,26,27,28,29,30] and experts’ opinion [31,32,33], we designed an exercise protocol aimed at improving quality of life (QoL) after THR and TKR for primary osteoarthritis. This goal is intended to be achieved by improving a variety of factors which influence QoL, including muscle strength, joint mobility, gait and balance function, and pain [13,14,24,25,26,27,28,29,30]. This paper presents the protocol of a single-blinded randomized trial aimed at evaluating the efficacy and safety of this exercise program. We hypothesize that after THR/TKR treatment those participating in this exercise program will have better outcomes in terms of quality of life and functional and clinical scores than persons following the usual care.

## 2. Materials and Methods

This study is carried out within the project “Physical ActIvity after hip and knee Replacement” (PAIR) and funded within the Erasmus Plus Sport program (Grant Agreement 613008-EPP-1-2019-1-IT-SPO-SCP). The study was approved by the Local Ethics Committee (Comitato Etico Indipendente di Area Vasta Emilia Centro, CE-AVEC) of the Emilia-Romagna Region (reference number AVEC: 1005/2020/Sper/IOR) and registered in ClinicalTrial.Gov (NCT04761367).

### 2.1. Study Design

The study is a randomized controlled trial. At the moment of the pre-surgery medical check-up, persons will be enrolled. After surgical treatment (THR or TKR) and subsequent rehabilitation treatment, persons will be randomly assigned to an intervention group or a control group. The intervention group will participate in a 6-month exercise program based on the PAIR exercise protocol and will receive educational sessions on the importance of maintaining an active lifestyle after THR or TKR. The control group will follow usual-care and will receive recommendations by surgeons and physiotherapist on the importance of maintaining an active lifestyle after THR or TKR. Six months after surgery, participants of both intervention and control group will be assessed before randomization (post-surgery baseline) and, subsequently, after 3 and 6 months (Figure 1).

### 2.2. Participant Recruitment

Study participation will be proposed to all patients during the pre-surgery clinical assessment (within 15 days before the scheduled day of surgical intervention) by the medical personnel of the Clinical Units Chirurgia Ortopedica Ricostruttiva Tecniche Innovative and Clinica Ortopedica e Traumatologica II of the Istituto Ortopedico Rizzoli of Bologna after verifying the presence of inclusion/exclusion criteria. Subsequently, patients will receive information about the study and will be requested to sign the informed consent form. Patients will be enrolled after signing the informed consent form.

### 2.3. Inclusion and Exclusion Procedures

During the pre-surgery assessment, the inclusion/exclusion criteria associated with age, diagnosis and comorbid conditions will be verified by a medical doctor, while criteria linked to motor function and physical activity by a professional trainer. Subsequently, criteria will be verified again at the post-surgery assessment. In this latter assessment will also be verified the presence of two additional inclusion criteria, i.e., post-surgery functional performance and pain (Table 1).

If eligible for the study, personal data (first and family name, address, telephone number(s)) will be recorded only on the informed consent form together with a study code. In all the other forms used in the study, patients will be identified with the patient’s study code.

### 2.4. Sample Size

The sample size was calculated through an a priori power analysis calculated on the study’s primary outcome, the physical health domain of the 36-item Short Form Health Survey (SF-36), assessed at post-surgery baseline and six-month assessments in both intervention and control groups. A randomized controlled pilot study in the literature with a similar rationale [36] was considered as a reference. Starting from this study, in which the SF-36 was used before and three months after the start of the intervention, was found a size effect of 1.06 (considered “increased”) with a standard deviation of 10 for each considered group. The power analysis was performed separately for THR and TKR, using a two-code test for independent groups with an error α = 0.05 and a sampling power (1-β) of 0.8, and a drop-out percentage of 30%. The minimum number required was estimated in 20 patients per group. Considering the four groups evaluated in the present study, two intervention groups (THR or TKR) and the two respective control groups, the final sample size was identified in a total of 80 patients.

### 2.5. Description of Procedure and Randomization

Both THR and TKR patients will be randomly assigned to the intervention groups and control groups after the post-surgery rehabilitation after having revised inclusion/exclusion criteria again. A randomization list, defined by a generator of random numbers available on the Emilia-Romagna Region website (http://wwwservizi.regione.emilia-romagna.it/generatore/, accessed on 11 April 2021) surgical intervention, 40 numbers within the interval (1,40) will be generated.

### 2.6. Allocation, Concealment and Blinding

This is a single blind RCT study. Random assignment to intervention groups or control groups will be performed by personnel other than those who will perform clinical and functional assessments. The list of allocation of the patients to the groups will be kept locked and separated from the rest of the material used to collect patients’ information. At any time, professionals who perform motor assessments will not be aware of which group participants have been assigned. Participants will be clearly instructed not to reveal the trainer who performs the assessment to which exercise group they have been assigned.

#### 2.6.1. Intervention Groups (IGs)

The intervention groups for THR and TKR will be trained under the direct supervision of a graduate trainer in Sports Science. The protocol aims to reach fitness by improving muscular strength, range of motion (ROM), and muscular elasticity, as well as gait, balance and aerobic capacity. The exercise program is structured in 2 days per week 1 h sessions, and lasts six months. Each session is divided into warm-up, strength, balance, flexibility and cool-down sections. The protocol defines strategies to instruct patients and check the correct and safe execution of the motor tasks for both THR and TKR. 

The exercises are chosen to avoid potentially harmful or painful movements or positions according to the participants’ conditions. For instance, exercises in quadrupedal position are excluded for persons with TKR. Exercises with leg adduction movements and in lateral decubitus positions are excluded for persons with THR. In order to follow the progressive workload adapting criteria, frequency, type, time and duration of the training will be modified based on the functional capacity reached by participants. Only common-use low-cost tools, such as elastic bands, yoga mats, dumbbells and soft dumbbells, are used within the exercise sessions. 

In addition, all persons of the intervention groups will be requested to choose an additional third day of the week to carry out at least one of the following activities: brisk walking, cycling or swimming. The duration of these additional activities will be at least 30 min in order to reach the minimum amount of exercise per week of 150/300 min recommended by WHO [23]. Participants to the intervention group will be monthly request how many hours of additional activities has been done during the last month.

#### 2.6.2. Control Groups (CGs)

The THR and TKR control groups will be requested to follow the usual care. The usual care is defined as the full spectrum of patient care practices in which clinicians have the opportunity to individualize care [37]. In this study, after having completed the rehabilitation treatment after surgery, patients will receive recommendations by surgeons and physiotherapists during follow-up visits on the importance and the appropriate quantity of weekly exercise in accordance with WHO recommendations [23]. Participants to the control group will be monthly contacted to monitor the compliance to the WHO’s recommendations. However, no structured long-term exercise training program will be provided.

### 2.7. Data Collection and Measures

All recruited participants will undergo multidimensional assessments before surgery and, subsequently, after completing the rehabilitation treatment (post-surgery baseline) and, 3 and 6 months later. Measures/records collected in each assessment session is summarized in Table 2 and pertain the following domains: general characteristics (age, gender, body mass index, comorbidity, education, social and professional), QoL, impairments and functional status, lifestyle, and safety. In addition, adherence to the exercise program, home-gym distance, and patient’s satisfaction will be investigated only in the participants of the THR and TKR intervention groups.

#### 2.7.1. General Characteristics

The factors that are considered relevant for the outcome of surgical treatment and/or physical activity programs are: age, gender, comorbidity, body mass index, educational level [38,39,40,41], social status [38,39,40], professional status [38,39,40], and home gym-distance [42], the possibility to reach the gym autonomously or with someone’s help, duration and number of sessions of rehabilitation treatment before and after surgery [43].

#### 2.7.2. Primary Outcome

The primary outcome is QoL’s modification, measured with the SF-36 [44,45,46]. SF-36 is one of the most widely used questionnaires to measure the health-related QoL in total hip and total knee arthroplasty patients [47]. SF-36 is composed of 36 items and is focused on measures of two main domains: physical and mental health. It is a multi-item scale that assesses eight health concepts [44]: (1) limitations in physical activities caused by health problems; (2) limitations in social activities because of physical or emotional problems; (3) limitations in usual role activities because of physical health problems; (4) body pain; (5) general mental health (psychological distress and well-being); (6) limitations in usual role activities because of emotional problems; (7) vitality (energy and fatigue); (8) general health perceptions. Validity and reliability of SF-36 have been previously proved also for the Italian version of the questionnaire [48].

#### 2.7.3. Secondary Outcomes 

Secondary outcomes are measures which may influence QoL.

Impairments, functional and clinical status.

Maximal strength of the lower limbs can be reliably measured with a hand-held dynamometer [49,50,51,52]. It measures the peak isometric force generated from a muscle group. It is widely used to evaluate the strength of knee [49,50,51] and hip muscles [52,53,54] in healthy elderly and people with osteoarthritis.

Hip and knee joints mobility [55] are measured by goniometry. 

The Hand Grip test will be measured by Hydraulic Hand Jamar Dynamometer. This measure has been strongly associated with frailty in the elderly population [56,57].

Pain is measured by visual analogue scale (VAS) [58,59,60]. VAS is a psychometric response scale widely used for measuring subjective characteristics that cannot be directly measured such as pain. It is widely used to measure pain in orthopedic studies.

Time Up and Go is a valid and reliable test used to examine physical performance and lower extremity [61]. It has been used in several clinical trials concerning different conditions [62,63,64], including arthroplasty [65,66,67,68]. It measures the time to stand from a standard chair, walk to a 3 m distance, turn around and return to sit on the same chair.

Single Stance test [69] is a useful tool for estimating standing balance and discriminating from low to high functional ability in individuals of various ages and functional levels. It is reliable, swift and easy to administer. It measures time subjects can balance on one leg up to a maximum of 30 s.

The 30 s Chair-Stand Test [70,71] is a valid and reliable single-item performance test for assessing lower limb function. It is performed by counting the number of stands completed in 30 s with hands crossed on the chest. 

The Harris Hip Score (HHS) [72] is a disease-specific, instrument widely used as outcome measure after THR. The domains covered are pain, function, absence of deformity and ROM. It has been translated and validated in the Italian language [73]. 

The Hip Disability and Osteoarthritis Score (HOOS) was developed in 2003 and showed to be effective in measuring patient-relevant outcomes in osteoarthritis patients even after THR [74]. It is a 39-item self-administered questionnaire with five separate sub-scales: pain, symptoms, stiffness, daily living activities, sport and recreation function, and hip-related quality of life. It has been translated and validated in several languages, including Italian [75].

The American Knee Society Scoring (KSS) [76] is a disease-specific scoring system developed to assess patients’ clinical outcomes and functional ability before and after total knee arthroplasty. Knees are examined for ROM, flexion contractures, extension leg, alignment and stability; functional score rates the patient’s ability to walk and climb stairs.

The Knee Injury and Osteoarthritis Outcome Score (KOOS) is a 42-item self-administered questionnaire that proved to be a reliable and valid instrument for evaluating surgery outcomes physical therapy [77,78] for knee impairments. It assesses five outcomes: pain, symptoms, daily living activities, sport and recreation function, and knee-related QoL. It has been translated and validated in Italian [79].

The Western Ontario and McMasters Universities Osteoarthritis Index (WOMAC) [80,81] is a valid and reliable instrument designed to provide a disease-specific measure for patients with osteoarthritis of hip and knee. It includes 24 items in three dimensions: pain (5 items), function (17 items) and stiffness (2 items). Validity and reliability of the instrument have also been demonstrated for the Italian version [82].

The High-Activity Arthroplasty Score (HAAS) [83,84] is focused on lower limb functions. It was specifically developed to assess functional ability variations after lower limb arthroplasty with particular regard to high functioning individuals. The score is a four-item self-assessment measure covering the four domains of walking, running, stair climbing, and general activities. The Italian version of the instrument has been previously validated [85].

##### Life Style

The Recent Physical Activity Questionnaire (RPAQ) [86] is a self-administered instrument. It is divided into three sections (home activities, activity at work, recreation) during the last four weeks. It has been translated and validated in different languages, including Italian [87]. Weekly activity questionnaires will also be used to monitor the physical activity performed by participants of the intervention group outside the gym and by those of the control group. Each participant will be requested to record the amount (minutes) of moderate/intense activity performed each week.

The PAIR questionnaire for patients’ attitudes toward physical activity is an instrument generated within the PAIR project. It investigates following domains: QoL, practice of physical activity, attitude towards physical activity, function, kinesiophobia.

##### Safety

Adverse clinical events (ACEs) that will occur to participants during the study will be carefully recorded. For the intervention groups, the trainer will record the ACEs occurred during the exercise sessions and outside the gym at the end of each session. In the case of three consecutive absences, the coordinating center will contact the participant by telephone to investigate whether the cause of non-attendance at the gym sessions was an ACE. For participants in control group, the ACEs will be registered by the study staff during follow-up. Based on the records, ACEs will be classified for severity (severe: if the ACE involved hospitalization/access to the emergency room; moderate: if the ACE required the intervention of a doctor and/or modification of the usual pharmacological therapy; mild: if the ACE did not require medical intervention and/or modification of the usual pharmacological therapy), place (home: ACE occurred at home; outside: ACE exercise occurred outside the home; gym: ACE occurred during the exercise session), and apparatus (apparatus/system involved).

##### Adherence

The adherence of each participant to the exercise program will be monitored in the intervention groups. The adherence will be measured as the per cent of exercise sessions actually performed/total number of scheduled exercise sessions. Moreover, also the third additional activity performed will be recorded.

##### Participants’ Satisfaction

Finally, the intervention group participants’ perception of the quality of the exercise program will be investigated, as it strongly influences the adherence to an exercise program [42]. It will be verified by a dedicated questionnaire with structured responses based on a 7-point Likert scale [88,89,90].

### 2.8. Statistical Analysis

The qualitative variables will be summarized in terms of frequency, and the quantitative ones in terms of mean and standard deviation for both intervention groups and control groups and for the three times of assessment. To compare characteristics between and within groups the principle of intention to treat will be used, adjusting for adherence to the exercise program. In order to compare the baseline characteristics between the two groups, the following tests will be used: Student’s t-test for parametric quantitative variables; Mann–Whitney test for non-parametric variables; Chi-square test for qualitative dichotomous variables. To compare the changes between and within intervention groups and control groups among baseline and follow-up assessments the following tests will be used: for the normally distributed variables, analyses of variance for repeated measures followed by t-test with Sidak post hoc corrections; for non-parametric variables, the Friedman test followed by the Wilcoxon test with Bonferroni correction.

## 3. Discussion

This protocol presents a randomized trial aimed at evaluating the efficacy and safety of an exercise program designed for improving the quality of life in people who had undergone THR or TKR. Most patients report successful long-term outcomes and reduced or cured pain after surgery [8,9,91], but recovery is variable, and the majority of patients continue to demonstrate lower extremity muscle weakness and functional deficits compared to age-matched control persons [92,93,94]. The exercise program could improve these limitations and prepared the persons to become active again in physical activity and sports.

Many studies have been focused on exercise after THR/TKR, but their goals differed from those of the present study. Indeed, the literature often focused on the correction of sarcopenia, neuromuscular function, and pain-induced disuse of the affected leg before and in the first weeks or months immediately after surgery [95,96,97]. Other studies focused on which type of athletic activities (e.g., golf, swimming, tennis, walking, mountaineering, jogging, tennis, gym activities) deemed acceptable in the long-term in order to avoid damages of the implanted arthroplasty [98,99,100]. Nevertheless, the study published regarding sport activity after THR and TKR, there are no recognized criteria for exercise prescription for those who have completed surgical and rehabilitative treatment after THR/TKR, and the workout characteristics are simply based on allowed and not recommended sports activities [31,32]. Thus, this study will investigate the safety and efficacy of an exercise program specifically designed for persons who have completed surgical and rehabilitative treatment after THR/TKR.

Physical activity has a primary role in preventing chronic diseases and their consequences associated with the aging process. Arthrosis is an aging process that leads to pain and, consequently, a state of inactivity. This problem has already been studied in the literature by programs tested in different countries, which aimed to increase physical activity, functional status and improve the response to the surgery [101]. In the case of severe osteoarthrosis, hip or knee replacement reduce pain and improve function; therefore, we would expect a significant change of motor behavior and hypothesis an active lifestyle by these persons [10] after surgery. However, most persons do not increase physical activity from pre-to-post-surgery but remain sedentary and present a high prevalence of diabetes, hypertension [13,17,18,100], and overweight or obesity [102]. Indeed, despite the total knee and hip prostheses lead to global benefits independently of body weight, the risk of disability increases among persons with high BMI following THR and TKR [103,104]; in particular, a higher obesity level correlates with lower post-surgery functional scores [105,106].

Adherence to an exercise program is a crucial aspect of successful training. This protocol includes gym group training because there is evidence that adherence is higher in supervised programs than in those unsupervised, as proved by systematic reviews [107,108,109,110,111]. Group training facilitates participation by reducing costs and promoting social interactions, which lead to improved social, mental, and emotional health [107,108].

Mobility and transport difficulties often preclude regular participation in an exercise program in senior citizens [27,42]. To solve this issue, we selected exercises that can be performed using easily available cheap equipment (elastic bands, yoga mats, dumbbells and soft dumbbells) instead of gym machines. This may facilitate the scaling up of the training program, if proved effective and safe, as it will be possible to replicate it not only in well-equipped gyms but also at home or in suitable spaces close to home. The use of community environments, where gyms are not available, has been reported to be very effective for sustains exercise adherence in previous research [42].

Health related QoL is considered of paramount importance to evaluate the impact of chronic diseases and treatment outcomes [112]. As a primary outcome measure, we selected the SF-36 because it is a multidimensional instrument widely used to evaluate the THR and TKR outcomes (see, e.g., References [113,114]). However, this is a generic instrument which has not been designed to approach QoL aspects specifically related to the evaluation of the outcomes of these surgical procedures. Thus, we included as secondary outcomes a multidimensional system of measures of impairments, functional and clinical status specifically related to the expected effects of the exercise program on the type of persons who are the focus of this study. This could allow a better understanding the results and, if necessary, further optimization of the exercise protocol.

### Limitation

The final goal of this study is to increase regular physical activity among the persons who underwent THR/TKR. Thus, to detect modifications of lifestyle activity represents an important goal of our project. However, to measure lifestyle activity is not an easy matter and several methods have been proposed including self-report questionnaires, smart phone technology, motion sensors, and pedometers [115]. All these methods have strengths and limitations in terms of methodological effectiveness and feasibility. The use of wearable devices or smart phones can be accurate but not easy to use in long-term study by aged participants. Thus, we decided to use a self-report questionnaire of physical activity (RPAQ) and weekly logs although social desirability may result in over-reporting of PA among participants keen to comply with the intervention aims [116]. These factors will require careful consideration in the interpretation of the results.

The protocol takes into account the duration and number of sessions of rehabilitation which participants have undergone before and after surgery, but not the type and intensity of this treatment. This may weaken the results of the study, as large and consistent evidence has proved the importance of pre- and post-surgery rehabilitation [43]. However, we deemed it unfeasible to reliably control this variable, as rehabilitation treatment will be performed not within the premises of our hospital but at the community level within private or public facilities. Nonetheless, differences in rehabilitation treatment outcomes observed at the post-surgery baseline will be used in the results’ analysis and interpretation.

We excluded people from participation in the study who are unable to walk unassisted for 500 m or have significant pain at rest. This may limit the participation of people who need the exercise the most and weaken the study conclusions. However, people with important functional limitations and/or severe pain are eligible for medical and rehabilitation care or aquatic exercise programs while land-based group exercise could be very problematic or even unsafe.

## 4. Conclusions

Exercise is fundamental to maintain or improve health and prevent disability in subjects who undergo THR/TKR. However, no specific indications on the appropriate exercise program are available for these subjects after the rehabilitation phase. The results of this study could add evidence for clinicians, exercise trainers, and policy makers toward the best strategy to ensure safe and effective exercise after surgery.

## Figures and Tables

**Figure 1 ijerph-18-06732-f001:**
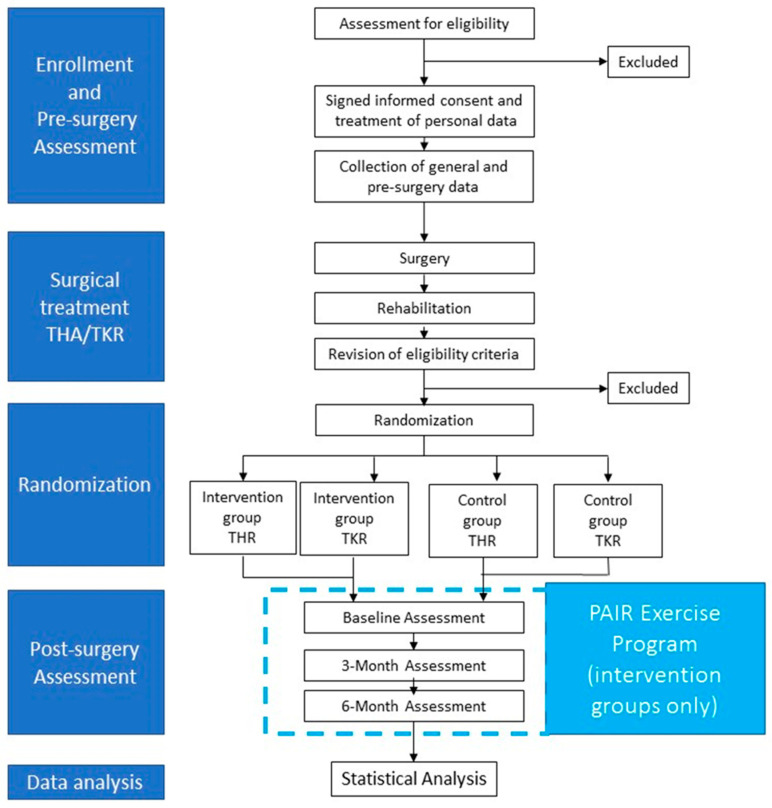
Study design.

**Table 1 ijerph-18-06732-t001:** Inclusion and exclusion criteria.

Inclusion Criteria	Exclusion Criteria	
Pre-Surgery Criteria	Pre-Surgery Criteria	Post-Surgery Criteria
Signed informed consent; Age: 50–80 years; Indications: Patient with advanced unilateral osteoarthritis requiring primary THR or TKR; General: ASA ^1^ class 1 or 2;	1. Unable/unwilling to sign the informed consent form of the study and/or willing to comply with the study requests; 2. Poor knowledge of Italian language which prevents understanding of the content of the consent form and/or of instructions for assessment and/or training; 3. Severe functional limitations of other lower extremity joints besides that for which surgery is planned; 4. Impairment of communicative and/or sensory functions so severe to make impossible understanding or executing trainer’s instructions (dementia, aphasia, blindness, deafness); 5. Heart failure (NYHA ^2^ class > 2); 6. Unstable angina; 7. Pulmonary disease requiring oxygen therapy; 8. Symptomatic peripheral arteriopathy; 9. Recent myocardial infarction or hospital admission for any other reason in the previous six months; 10. Symptomatic orthostatic hypotension; 11. Hypertension in poor pharmacologic control (diastolic > 95 mmHg, systolic > 160 mmHg); 12. Relevant neurological condition impairing motor or cognitive function; 13. Any other condition that the medical doctor considers contraindicating the participation in an exercise program of moderate intensity; 14. Severe depression	Functional performance: Able to stand and walk > 500 m independently; Pain: score ≤ 4 in VAS ^3^ during rest.

^1^ ASA: American Society of Anesthesiologists; ^2^ NYHA: New York Heart Association Functional Classification; ^3^ VAS: Visual Analogue Scale.

**Table 2 ijerph-18-06732-t002:** Variables collected during the study and time of recording.

	Assessments			
	before Surgery	after Surgery and Rehab		
		Baseline	3 Months	6 Months
General characteristics				
Age	X			
Gender	X			
Body mass index	X			
Comorbid conditions	X			
Education	X			
Social status	X			
Professional status	X			
Home-Gym Distance (only for intervention groups) Possibility to reach the gym in autonomy		X X		
Rehabilitation treatment (duration and number of sessions)		X		
Rehabilitation treatment				
Before surgery (duration and number of sessions)		X		
After surgery (duration and number of sessions)		X		
Primary outcome				
Quality of life				
Short-Form Health Survey (SF-36)	X	X	X	X
Secondary outcomes				
Impairments, functional and clinical status				
Maximal strength of the lower limb	X	X	X	X
Hip and knee mobility	X	X	X	X
Hand Grip test	X	X	X	X
Visual Analogue Score (VAS)-Pain	X	X	X	X
Time Up and GO (TUG)	X	X	X	X
Single Stance test		X	X	X
30 s Chair-Stand Test (30 s-CST)		X	X	X
Harris Hip Score (HHS) *	X	X	X	X
Hip disability and Osteoarthritis Score (HOOS) *	X	X	X	X
The American Knee Society scoring (KSS) **	X	X	X	X
Knee Injury and Osteoarthritis Outcome Score (KOOS) **	X	X	X	X
The Western Ontario and McMasters Universities Osteoarthritis Index (WOMAC)	X	X	X	X
The High-Activity Arthroplasty Score (HAAS)	X	X	X	X
Lifestyle				
Recent Physical Activity Questionnaire (RPAQ)	X	X	X	X
Weekly physical activity logs		During whole exercise intervention		
PAIR questionnaire for patients’ attitudes toward physical activity		X	X	X
Safety				
Adverse clinical events		X	X	X
Adherence to the exercise program				
Adherence to the exercise program (only for the intervention groups)		During whole exercise intervention		
Patients’ satisfaction (only for the intervention groups)				
Questionnaire on satisfaction of the exercise program				X

* for THR only; ** for TKR only.

## Data Availability

No new data were created or analyzed in this study. Data sharing is not applicable to this article.

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
