# Peer review of "Evaluation of the Efficacy and Safety of an Exercise Program for Persons with Total Hip or Total Knee Replacement: Study Protocol for a Randomized Controlled Trial"

_ijerph, 2021, doi:10.3390/ijerph18136732_

Round 1
Reviewer 1 Report
Thank you for the opportunity to review your intervention protocol. I agree with the authors that there is a need to study ways of increasing for long-term physical activity levels post THR/ TKA. Here are a few suggestions or factors to consider
- Is it necessary to use the acronyms: PA, IG, CG?
- Can physical activity also be objectively monitored using activity trackers (smart watches) and/or participant journals (activity diary/ logs)? This will ensure that we can track the level of physical activity that participants in both groups and how that is associated with QOL.
- I agree with the logic of using simple home gym type equipment to ensure carryover following the intervention in the intervention group. But how that will address the transportation barrier? Am not sure what steps are being taken to ensure that study grp participants will be able to come to the intervention site during the study and if they will be connected to local community gyms after?
- Have you considered measuring environmental factors, including family support that hinder physical activity and hence QOL?
- An environmental frame of reference or a broader QOL assessment is needed to increase our understanding of factors that contribute to physical activity post THR/TKR.
Author Response
Dear Reviewer,
We are very grateful to the referees for your comments and criticism and reviewed the manuscript in accordance.
Please find below the modifications we made.
Best regards,
Giuseppe Barone
Comments and Suggestions for Authors
Thank you for the opportunity to review your intervention protocol. I agree with the authors that there is a need to study ways of increasing for long-term physical activity levels post THR/ TKA. Here are a few suggestions or factors to consider
R1C1: Is it necessary to use the acronyms: PA, IG, CG?
A: We removed all these acronyms in the text.
R1C2: Can physical activity also be objectively monitored using activity trackers (smart watches) and/or participant journals (activity diary/ logs)? This will ensure that we can track the level of physical activity that participants in both groups and how that is associated with QOL.
A: We agree with the reviewer point of view since the increase of active behavior after THR and TKR is the final goal of our project. Thus, in addition to RPAQ, we introduced a weekly log to record the modification of physical activity of all participants “life style” of methods. We added the sentence: “Weekly activity questionnaires will also be used to monitor the physical activity performed by participants of the intervention group outside the gym and by those of the control group. Each participant will be requested to record the amount (minutes) of moderate/intense activity performed each week.”. We modified Table 2 accordingly (L.402). Moreover, in limitation section fo the discussion we added the following paragraph: “The final goal of this study is to increase regular physical activity among the persons who underwent THR/TKR. Thus, to detect modifications of lifestyle activity represents an important goal of our project. However, to measure lifestyle activity is not an easy matter and several methods have been proposed including self-report questionnaires, smart phone technology, motion sensors, and pedometers [115]. All these methods have strengths and limitations in terms of methodological effectiveness and feasibility. The use of wearable devices or smart phones can be accurate but not easy to use in long-term study by aged participants. Thus, we decided to use a self-report questionnaire of physical activity (RPAQ) and weekly logs although social desirability may result in over-reporting of PA among participants keen to comply with the intervention aims [116]. These factors will require careful consideration in the interpretation of the results.” (L.519)
R1C3: I agree with the logic of using simple home gym type equipment to ensure carryover following the intervention in the intervention group. But how that will address the transportation barrier? Am not sure what steps are being taken to ensure that study grp participants will be able to come to the intervention site during the study and if they will be connected to local community gyms after?
A: We agree with the issues raised by the reviewer. We believe that the strategy of using simple equipment could be useful to perform physical activity in convenient adapted spaces where gyms are not easily available, such as home, parish and social clubs. We added the following sentence in discussion: “The use of community environments, where gyms are no available, has been reported to be very effective for sustains exercise adherence in previous research.” (L.504).
R1C4: Have you considered measuring environmental factors, including family support that hinder physical activity and hence QOL?
A: We agree with the reviewer, social and environmental status are already present in “general characteristics” as well as family status in this protocol.
R1C5: An environmental frame of reference or a broader QOL assessment is needed to increase our understanding of factors that contribute to physical activity post THR/TKR.
A: We agree with the reviewer. In order to extend QoL assessment, we added other parameters, such as “possibility to reach the gym in autonomy” and “weekly physical activity logs” (Tab. 2) (L.323)
Reviewer 2 Report
The manuscript presents a very interesting and timely protocol for RCT evaluating exercise program after TKA and THA. My overall opinion is that this paper is well written and structured containing comprehensive information about the design of a future RCT. Furthermore, it is approved by proper EC and registered as a trial in a database to avoid redundancy.
My comments are as follows:
Introduction:
The first sentence is supported by 7 references which is rather odd. Those that do not add any relevant information may be omitted.
The authors have stated that “Regular Physical Activity (PA) is one of the most effective interventions to improve the prevalent chronic comorbid conditions… [12]". Importantly, it is one of the strongest modifiable risk factors for knee and hip OA by inducing quadriceps muscle hypertrophy and improve strength in patients with weakness and atrophy related to knee pathology while appearing safe when properly performed [PMID: 30911813]
Methods:
PICO framework is followed.
Validated PRO and objective outcomes are planned and will be implicated in the follow-up. Primary and secondary outcomes are well defined.
I have nothing relevant to add to the methods and discussion part.
Overall, an excellent paper describing a protocol for RCT. Hope it will be accomplishable in real-world settings.
Author Response
Dear Reviewer,
We are very grateful to the referees for your comments and criticism and reviewed the manuscript in accordance.
Please find below the modifications we made.
Best regards,
Giuseppe Barone
Comments and Suggestions for Authors
The manuscript presents a very interesting and timely protocol for RCT evaluating exercise program after TKA and THA. My overall opinion is that this paper is well written and structured containing comprehensive information about the design of a future RCT. Furthermore, it is approved by proper EC and registered as a trial in a database to avoid redundancy.
My comments are as follows:
Introduction:
R2C1: The first sentence is supported by 7 references which is rather odd. Those that do not add any relevant information may be omitted.
A: We removed three references focused on socioeconomic burden of TKA-TKR. (L.57)
R2C2: The authors have stated that “Regular Physical Activity (PA) is one of the most effective interventions to improve the prevalent chronic comorbid conditions… [12]". Importantly, it is one of the strongest modifiable risk factors for knee and hip OA by inducing quadriceps muscle hypertrophy and improve strength in patients with weakness and atrophy related to knee pathology while appearing safe when properly performed [PMID: 30911813]
A: We added a sentence and a citation as suggested by the reviewer: “Furthermore, since physical activity induces quadriceps hypertrophy and improves strength, it is one of the main modifiable factors in patients with knee and hip osteoarthrosis who often exhibit weakness and atrophy” (L.72) and introduce references 12.
Methods:
PICO framework is followed.
Validated PRO and objective outcomes are planned and will be implicated in the follow-up. Primary and secondary outcomes are well defined.
I have nothing relevant to add to the methods and discussion part.
Overall, an excellent paper describing a protocol for RCT. Hope it will be accomplishable in real-world settings.
Reviewer 3 Report
I have reviewed the submission by Barone et al regarding a RCT involving a somewhat personalized exercise protocol to be used on patients receiving either a total hip or knee replacement via conventional surgery during the post-surgery interval from rehab for 6 months. A matching Control cohort will receive standard care. Patients meeting the inclusion criteria will be assessed at baseline and at 3 & 6 months post-surgery. Exclusion of post-TJR patients will be those experiencing excess pain or unable to walk 500 meters unassisted.
The literature would certainly support the need to enhance the physical activity of post-TJR patients to increase their quality of life and further enhance the gains in functional improvement in mobility. Many protocols have been tried and the outcomes are quite variable, primarily due to failure to continue the exercises when not supervised. Therefore, the problem is an important one, but the protocol is not unique.
In addition to the lack of uniqueness, there are other ocncerns with the manuscript as presented:
- After a TJR surgery, ~30% of those receiving such surgery do not experience relief from pain. Thus, their pain is still quite severe. This population would likely be excluded from the protocol. The criteria of being able to walk unassisted for 500 meters after initial rehab would likely select for those that are high achievers and bias against those that may actually need the exercise more acutely.
- There was no mention of balance between males and females in both the Control and Intervention cohorts.
- Many Patients with advanced OA are in a deconditioned state due to their functional disabilities, a condition further exacerbated by the surgery and the immediate rehab period. Therefore, there is a considerable body of literature that has argued for "pre-hab" exercise protocols, including the GLA:D program from Denmark, to alleviate the pre-surgery deconditioned state and thus enhance the response to surgery and set the stage for the post-surgery environment. This was not discussed.
- It may be difficult to monitor the compliance of patients performing the unsupervised part of the protocol. Patient compliance is a consistant and on-going problem with this patient population which is mainly older, often more frail, and satisfied with not having as much pain, and less worried about meeting the researchers expectations.
- The authors are encouraged to insert a "Limitations" section into the Discussion.
- Why do the authors use the SF-36 as the only primary outcome?
Author Response
Dear Reviewer,
We are very grateful to the referees for your comments and criticism and reviewed the manuscript in accordance.
Please find below the modifications we made.
Best regards,
Giuseppe Barone
Comments and Suggestions for Authors
I have reviewed the submission by Barone et al regarding a RCT involving a somewhat personalized exercise protocol to be used on patients receiving either a total hip or knee replacement via conventional surgery during the post-surgery interval from rehab for 6 months. A matching Control cohort will receive standard care. Patients meeting the inclusion criteria will be assessed at baseline and at 3 & 6 months post-surgery. Exclusion of post-TJR patients will be those experiencing excess pain or unable to walk 500 meters unassisted.
The literature would certainly support the need to enhance the physical activity of post-TJR patients to increase their quality of life and further enhance the gains in functional improvement in mobility. Many protocols have been tried and the outcomes are quite variable, primarily due to failure to continue the exercises when not supervised. Therefore, the problem is an important one, but the protocol is not unique.
In addition to the lack of uniqueness, there are other concerns with the manuscript as presented:
R3C1: After a TJR surgery, ~30% of those receiving such surgery do not experience relief from pain. Thus, their pain is still quite severe. This population would likely be excluded from the protocol. The criteria of being able to walk unassisted for 500 meters after initial rehab would likely select for those that are high achievers and bias against those that may actually need the exercise more acutely.
A: We gave a lot of attention to the exclusion criteria in collaboration with orthopedic surgeons. Our main concern was to exclude people unable to walk unassisted for 500 meters and pain at rest since it may limit the participation of people who need the exercise the most. However, people with important limitations or still in severe pain are eligible for other kind of medical treatments or PA programs (aquatic). In addition, we had great concern about safety. Thus, in the Limitations section of the Discussion we better explained the reasons of these exclusion criteria. “We exclude form the participation of the study people who are unable to walk unassisted for 500 meters or have significant pain at rest. This may limit the participation of people who need the exercise the most and weakens study conclusions. However, people with important functional limitations and/or severe pain are eligible for medical and rehabilitation care or aquatic exercise programs while land-based group exercise could be very problematic or even unsafe.”
R3C2: There was no mention of balance between males and females in both the Control and Intervention cohorts.
A: The study was not aimed to analyze difference due to gender. However, with the randomization process, since no preference will be given to either gender, but patients will be offered the opportunity to participate based on surgery waiting lists, we expect to have percentages of male and female which reflects gender characteristics of the patients who undergo THR/TKR (approximately THR 55% female and 45% male, TKR 65% female and 35% male) in Control and Intervention groups.
R3C3: Many Patients with advanced OA are in a deconditioned state due to their functional disabilities, a condition further exacerbated by the surgery and the immediate rehab period. Therefore, there is a considerable body of literature that has argued for "pre-hab" exercise protocols, including the GLA:D program from Denmark, to alleviate the pre-surgery deconditioned state and thus enhance the response to surgery and set the stage for the post-surgery environment. This was not discussed.
A: The long-term outcomes may be influenced by physical activity before surgery, length of rehabilitation program etc.. Given that we recorded all the physical activity and rehabilitation pre-surgery and post-surgery. Moreover, as the reviewer suggests, in order to argue of physical activity pre-surgery, we added the following sentences:” Arthrosis is an ageing process that leads to pain and, consequently, a state of inactivity. This problem has already been studied in literature by programs tested in different countries, which aimed to increase physical activity, functional status and improve the response to the surgery” (L.481)
In addition, we modified the following sentence as: “In case of severe osteoarthrosis, hip or knee replacement…” (L.484).
R3C4: It may be difficult to monitor the compliance of patients performing the unsupervised part of the protocol. Patient compliance is a consistant and on-going problem with this patient population which is mainly older, often more frail, and satisfied with not having as much pain, and less worried about meeting the researchers expectations.
A: We added logs (see answer Referee 1) to record physical activity in both groups. We agree with the difficult to monitor the compliance of patients performing physical activity. Therefore, we scheduled monthly contacts to the participants of both groups to monitor and sustain participants’ compliance to the WHO’s recommendations (L.302). We added the following sentences into the Materials and Methods: “Participants to the intervention group will be monthly request how many hours of additional activities has been done during the last month.” ; “Participants to the control group will be monthly contacted to monitor the compliance to the WHO’s recommendations.”
R3C5: The authors are encouraged to insert a "Limitations" section into the Discussion.
A: Thank you for the comment. We added the section named “Limitations” in the Discussion as suggested by the reviewer (L.518).
R3C6: Why do the authors use the SF-36 as the only primary outcome?
A: We agree that there are different important aspects to consider when dealing with quality of life. We selected SF36 as primary outcome measure since widely used in published literature also to evaluate QoL after THR and TKR. Nonetheless we designed a multidimensional assessment of all participants to include different aspects of functioning which may influence QoL.
Round 2
Reviewer 3 Report
The authors have addressed my major concerns and modified the manuscript appropriately. The study will have some significant limitations, but these are now addressed in the Discussion.